# Segmentation strategy of de novo designed four-helical bundles expands protein oligomerization modalities for cell regulation

Estera Merljak [1,2], Benjamin Malovrh [1] & Roman Jerala [1] ✉

Protein–protein interactions govern most biological processes. New protein assemblies can be introduced through the fusion of selected proteins with di/oligomerization domains, which interact specifically with their partners but not with other cellular proteins. While four-helical bundle proteins (4HB) have typically been assembled from two segments, each comprising two helices, here we show that they can be efficiently segmented in various ways, expanding the number of combinations generated from a single 4HB. We implement a segmentation strategy of 4HB to design two-, three-, or four-chain combinations for the recruitment of multiple protein components. Different segmentations provide new insight into the role of individual helices for 4HB assembly. We evaluate 4HB segmentations for potential use in mammalian cells for the reconstitution of a protein reporter, transcriptional activation, and inducible 4HB assembly. Furthermore, the implementation of trimerization is demonstrated as a modular chimeric antigen receptor for the recognition of multiple cancer antigens.

Synthetic biology strives toward constructing complex systems governed by specific, designed protein–protein interactions that do not interfere with other components. One solution that avoids large protein modifications is the fusion of proteins of interest with oligomerization domains. Researchers have investigated dimerization domains, which spontaneously interact with their designed partner. When genetically fused to non-interacting proteins, dimerization domains serve as a tool for bringing the two fused proteins into close proximity[1–3]. To decrease non-desired interference with other components and have a low genetic footprint, small oligomerization domains are preferred. Helical bundles are a good example of such oligomerization domains. They consist of varying numbers of helices and are frequently found among natural proteins[4]. A subset of such structures is four-helical bundles (4HBs), which play important roles in the mammalian immune system[5], formation of cell contacts[6], and bacterial chemotaxis[7]. The general 4HB structure has been extensively studied[8–10], which facilitates the re-design of naturally occurring 4HBs[11,12] and determines the rules for de novo design of synthetic 4HBs[13–15]. Rational design is a potent tool for constructing synthetic

proteins that do not interfere with natural cell processes. De novo designed helical bundles have been used as regulators of protein interactions in synthetic transcription systems[1–3,16], interaction scaffolds in engineered metabolic pathways[17], and structures for binding specific ions and molecules[18–20], ion channels[21], and biomaterials[22–24]. However, sets of orthogonal 4HBs are likely to have a limited number of members due to design restrictions within the layers of 4HBs[25]. The Rosetta-based HBNet[26] design of hydrogen-bond networks inside the core of helical bundles introduces complex interaction surfaces with greater interaction specificity within the sets of orthogonally designed structures[25,26]. Orthogonal sets of heterodimers forming 4HBs have been previously characterized both in vitro and in situ for designing protein logic gates[27] and as surface antigen recognition modules[28].

In this study, we explore the potential of expanding the orthogonal set of interacting domains based on 4HBs by implementing a segmentation strategy of a single 4HB, creating a set of diverse oligomerization modules. In principle, 4HBs can be split into various subdomains by combining two to four interacting modules, which can reconstitute a 4HB assembly. In addition to expanding the set of

[1]Department of Synthetic Biology and Immunology, National Institute of Chemistry, Ljubljana, Slovenia. [2]Interdisciplinary Doctoral Programme of Biomedicine, Faculty of Medicine, University of Ljubljana, Ljubljana, Slovenia. ✉e-mail: roman.jerala@ki.si

heterodimers, it would be beneficial to generate heterotrimerizing and tetramerizing 4HB-based modules.

Here we prepared different oligomerization domains by segmentation of a single designed 4HB (DHD13_XAAA)[25] into di-, tri-, and tetramerization modules and applied hetero-oligomerization domains for the regulation of mammalian cell-signal recognition and processing, design of synthetic pathways and logic gates, and formation of a synthetic multiCAR-T technology platform for cellular therapy. Considering the high modularity of helical bundles, the presented segmentation technology can be applied for diverse biological and medical applications.

## Results

### Design of split4HB oligomerization domains

Four-helical bundles (4HB) are protein structures comprising four helices that assemble into a 3D topology based on helix–helix interactions (Fig. 1a). Most of the natural single-chain 4HBs are antiparallel, composed of up-down–up-down-linked segments. When fused with other proteins, HB-forming peptides can bring together proteins that do not have intrinsic affinity and thus act as oligomerization domains[27]. The compositions of 4HBs from the four modules enable a larger number of combinations compared to other oligomerizing domains, such as protein heterodimers or coiled-coil dimers. So far, 4HB segmentation has mainly been performed for a 2:2 combination of helices, where homodimeric interaction is often encoded[25–27].

We aimed to prepare different combinations of oligomerization domains through the permutation of backbone connectivity and splitting within the linker regions of a designed 4HB composed of four distinct helical domains. A set of dimerization, trimerization, and tetramerization domains was constructed from a single 4HB. Dimerization can be accomplished by segmentation into 2:2 or 1:3 modules, trimerization from 2:1:1, and tetramerization from a 1:1:1:1 combination (Fig. 1b). Thus, in principle, there are six possible dimerization combinations (two for 2:2 and four for 1:3 modules), four trimerization combinations, and a single tetramerization combination from a single 4HB. Many of these oligomerization segmentations may be orthogonal, which substantially increases the number of oligomerization combinations that can be applied for various purposes in synthetic biology.

To test the reconstitution of split4HB combinations from multiple modules, we designed constructs containing different segmentations

of 4HB peptides as fusion proteins using either an N- or a C-terminal split firefly luciferase domain as a reporter protein. The interaction of peptides and reassembly of the 4HB structure brings the two split luciferase domains into close proximity, reconstituting firefly luciferases' enzymatic activity (Fig. 2a). Since most applications of interest are likely to occur in cells, we performed these tests in mammalian cells.

Dimerization domains can be constructed as semi-equal (2:2) or unequal (1:3) segmentations of 4HBs. The arrangement of peptides is denoted in the names based on the number of peptide segments (e.g., AB:CD and DAB:C).

Semi-equal dimerization segmentation comprises two genetically fused peptides in each dimerization module (Fig. 1b). This reduces the number of combinations for the antiparallel 4HB due to peptide orientation, as only two combinations of antiparallel segments can be fused by a short linker, allowing the fusion of AB and DA combinations with CD and BC, respectively. We observed successful 2:2 dimerization of both AB:CD and DA:BC 2:2 arrangements (Fig. 2b), which was not significantly affected by which split luciferase domain was fused to one or the other dimerization module (Supplementary Fig. 1a). Unequal segmentation results in dimerization modules with one peptide in one and three peptides in the other dimerization domain (Fig. 1b). In this case, there are four possible combinations, regardless of the 4HB topology. Different levels of dimerization were observed among the different dimerization domains. A:BCD, BCD:A, B:CDA, and CDA:B resulted in strong; D:ABC and ABC:D in medium; and C:DAB, and DAB:C in a low reconstitution of the split luciferase (Fig. 2c; Supplementary Fig. 1b).

Observed differences were unexpected, as the interaction surfaces of all four peptides were similar and expected to result in a comparable affinity. These results indicate that, with segmentation into 1:3 dimerization modules, the contribution of individual peptides to 4HB formation can be evaluated. Therefore, for this particular 4HB, three of the four theoretically possible 1:3 combinations were successful for 4HB assembly.

### Orthogonality of 2:2 and 1:3 dimerization domains

Next, we evaluated the interactions between different segmentations of the same 4HB (Fig. 3a). The implementation of different segmentations of the same 4HB without cross-talk can expand the number of multiplexable partner proteins. Therefore, we tested orthogonality

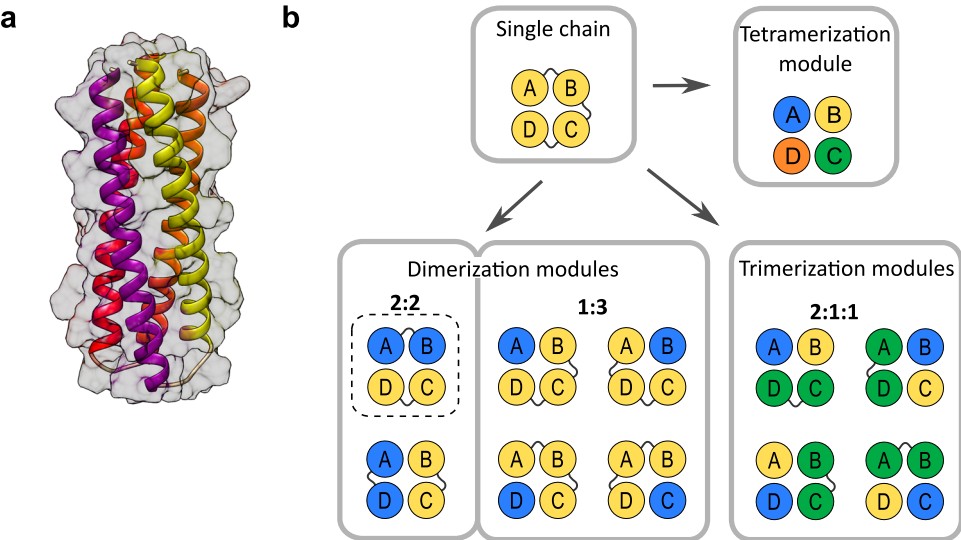

**Fig. 1 | Four-helical bundle segmentation strategy for the generation of multiple oligomerization modules. a** Crystal structure of DHD13_XAAA 4HB determined by NMR (PDB:6DMP). **b** Segmentation strategy for 4HB by permutation of backbone connectivity and splitting within the linker regions. The hatched line represents the conventional design.

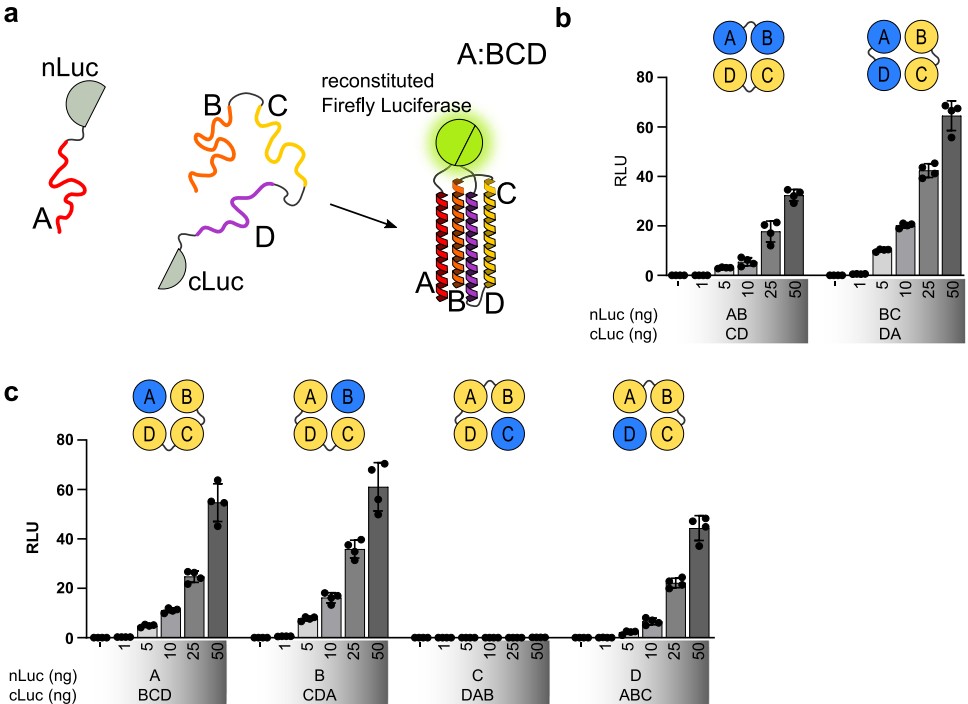

**Fig. 2 | Design and characterization of 4HB dimerization domains in HEK293T cells. a** Schematic of the split firefly luciferase system explained on A:BCD dimerization of 4HB. 4HB is split into two parts−peptide A and peptides BCD −and fused with the N-terminal (nLuc) and C-terminal (cLuc) domains of firefly luciferase, respectively. Peptides are unstructured until in close enough proximity for the formation of a 4HB, where each peptide structures into an alpha helix. The 4HB formation brings the two parts of split luciferase in spatial proximity sufficient for reconstitution and, thus, active firefly luciferase. **b**−**c** Evaluation of designed split4HB domains in HEK293T for reconstitution efficiency and concentration dependence of 2:2 dimerization domains (**b**) and 1:3 dimerization domains (**c**). Values in (**b**−**c**) are the mean of four biological replicates ± (s.d.) and representative of three independent experiments.

among all six possible dimerization modules. While a weak tendency for homodimerization was present in 2:2 segmentations, it was at least twice weaker than the designed dimerization for DA and even less for AB and BC dimerization domains. With 1:3 dimerization domains, we also observed some cross-talk; however, the designed dimerization pairs were favorable in all cases (Fig. 3b). Out of the six possible dimerization combinations, three proved mutual orthogonality, namely A:BCD, D:ABC, and AB:CD, regardless of the arrangements of the split luciferase domains. Alternatively, also combinations of B:CDA and AB:CD demonstrated high orthogonality. The orthogonality of different 2:2 and 1:3 combinations demonstrates that several segmentation combinations from a single 4HB protein can be used simultaneously; however, the appropriate module combinations, tailored for each 4HB of interest, should be selected with caution.

To prove the possible simultaneous use of different 4HB dimerization domains for transcriptional activation, we designed and tested a two-reporter expression system. The first module included a DNA-binding protein TALE[F] fused to peptide A, a BCD module fused to a transcriptional activator VP16, and a reporter plasmid encoding a fluorescent protein mCitrine under a minimal promoter ($p_{min}$) and TALE[F]-binding sites. The second module included a DNA-binding protein TALE[E] fused to peptide D, an ABC segment fused to VP16, and a reporter encoding a fluorescent protein tagBFP under a minimal promoter (pmin) and TALE[E]-binding sites. A:BCD dimerization resulted in mCitrine expression, while D:ABC dimerization resulted in tagBFP expression (Fig. 3c). With the combination of the above-mentioned constructs, we could also evaluate the cross-talk between the two investigated segmentations. The flow cytometry results confirmed the orthogonality of A:BCD and D:ABC segmentations of the same 4HB and thus the possibility of simultaneous use of two 1:3 segmentations without cross-talk (Fig. 3c, Supplementary Fig. 2). Taken together, these results support our hypothesis of using segmentation

as a way to generate multiple orthogonal oligomerization domains from a single 4HB.

## Protease-inducible dimerization of 1:3 segmentation domains

To achieve a fast shift from state A to state B, cells have "ready-to-go" systems in which all components are already present−requiring only a trigger signal to induce a selected process, which can be designed based on the protein interaction or modification[29–31]. To construct such a system, we designed an inducible dimerization of the 4HB assembly, where the interaction of the two partners, A and BCD, was prevented due to the presence of an inhibitory module B attached to the A module. This inhibition can be relieved by proteolytic cleavage of the linker between the A and B modules (AsB), which comprise the cleavage site for a selected protease. Upon cleavage with tobacco etch virus protease (TEVp) or Southern bean mosaic virus protease (SbMVp), peptide B is cleaved off peptide A, which allows A:BCD dimerization and reconstitution of a split luciferase reporter (Fig. 4a). First, we established a concentration dependence of TEVp- and SbMVp-inducible and non-inducible dimerization in the absence and presence of TEVp (Fig. 4b) or SbMVp (Fig. 4c). We found that the fusion of *tevs* or *sbmvs* and peptide B to the nLuc_A (AsB) construct indeed prevented dimerization in the absence of either TEVp or SbMVp (Fig. 4b, c). The assessed inhibitory efficiency of the fused peptide B on dimerization in the absence of a protease at 10 ng of plasmids encoding TEVp- and SbMVp-inducible dimerization domains was 64- and 82-fold, respectively (no protease (−) in Supplementary Fig. 3a, b). The co-expression of TEVp or SbMVp resulted in up to 45- and 70-fold increases in A:BCD dimerization and subsequent luciferase activity for TEVp- and SbMVp-inducible dimerization domains, respectively (Fig. 4b, c).

The TEVp- and SbMVp-inducible dimerization domains demonstrated concentration-dependent activation; however, a slight

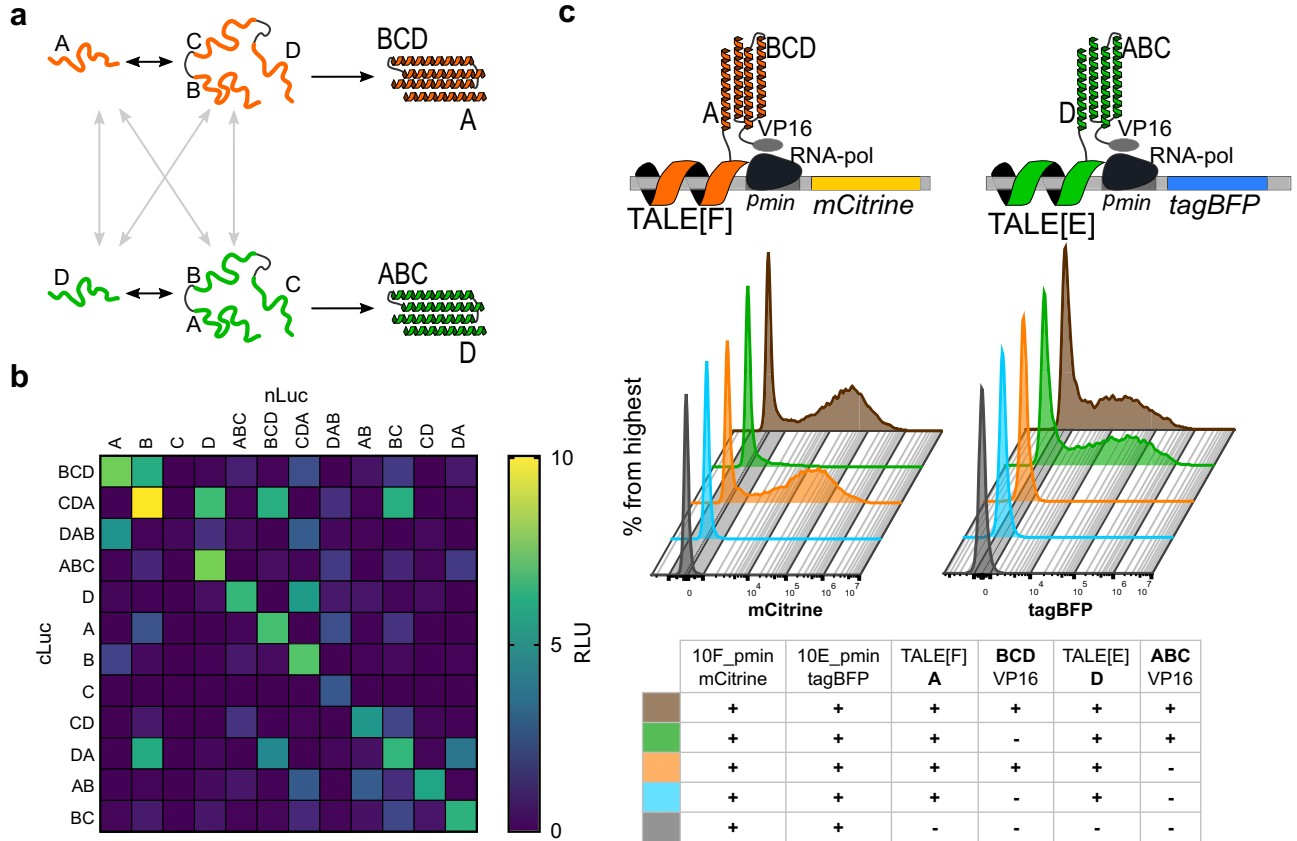

**Fig. 3 | Orthogonality of 2:2 and 1:3 segmented 4HB dimerization modules in HEK293T cells. a** Schematic of orthogonality experiment design. Two 1:3 segmentations interact with their designed partner (black arrows), successfully forming a 4HB; interactions between modules from different segmentations are not desired (gray arrows). **b** Orthogonality of the designed 2:2 and 1:3 dimerization domains. HEK293T cells were transiently transfected with 25 ng of the denoted dimerization constructs. **c** Simultaneous use of A:BCD and D:ABC dimerization modules in HEK293T cells evaluated by flow cytometry. HEK293T cells were transiently transfected with a combination of TALE reporter plasmids, DNA-binding TALE fused with one segment of 4HB, and transcriptional activators fused with the

counterparts of 4HB dimerization modules, as noted in the legend. The reconstitution of A:BCD and D:ABC leads to mCitrine and tagBFP expression, respectively. Front to back: reporter plasmids only (gray line); reporter plasmids, TALE[F]-A and TALE[E]-D (blue line); reporter plasmids, TALE[F]-A, TALE[E]-D, and BCD:VP16 activator domain (orange line; expecting mCitrine expression); reporter plasmids, TALE[F]-A, TALE[E]-D, and ABC:VP16 activator domain (green line; expecting tagBFP expression); reporter plasmids, TALE[F]-A, TALE[E]-D, BCD:VP16, and ABC:VP16 (brown line; expecting mCitrine and tagBFP expression). Values in (**b**) are the mean of four biological replicates and representative of three independent experiments. Graphs in (**c**) are representative of two independent experiments.

decrease in the luciferase activity of non-inducible dimerization domains was observed at higher amounts of either TEVp or SbMVp, which can be attributable to the nonspecific protease cleavage (Supplementary Fig. 3a, b).

With the aim of external chemical regulation of protease-inducible dimerization systems, we used a previously reported splitTEVp-FRB and FKBP domain fusion[29], where the protease reconstitution is activated by a chemical inducer rapamycin or its rapalog derivative. The addition of rapalog dimerizes FRB and FKBP fused to splitTEVp domains, resulting in activated splitTEVp, which cleaves the inhibitory peptide B and enables the dimerization of the segmented 4HB (Fig. 4d). Titration of splitTEVp domains was performed at constant amounts of AsB and BCD dimerization domains, where we observed a clear correlation between the increasing amounts of splitTEVp domains and reconstitution of TEVp-inducible dimerization domains in the presence of rapalog (Supplementary Fig. 3c). We assessed the kinetics of rapalog induction of TEVp-inducible dimerization in the absence and presence of 1 μM rapalog (Fig. 4e), as well as its efficiency, by comparing TEVp-inducible and non-inducible dimerization signals (Supplementary Fig. 3d) at a series of time points. Even with short induction times, we could observe a statistically significant increase in the TEVp-inducible dimerization signal in the presence of 1 μM rapalog (Fig. 4e, Supplementary Table 1).

To test the ability of the segmented 4HB platform to construct regulatory circuits, we designed a protease cascade from the segmented 4HB for the reconstruction of split plum pox potyvirus protease (splitPPVp) activity by controlling the association of the splitPPVp domains. For this, we used a set of orthogonal proteases, as described previously[29]. First, we determined the efficiency of splitPPVp reconstitution as the N- and C-terminal fusions with segmented A:BCD 4HB dimerization domains. The best combinations were the fusion of N-splitPPVp with peptide A (nPPVp_A) and that of C-splitPPVp with the BCD module (BCD_cPPVp) (Supplementary Fig. 4a). Based on these results, we prepared constructs from SbMVp-regulated 4HB dimerization domains, where the split luciferase domains were substituted with the splitPPVp domains (nPPVp_A:sbmvs:B and BCD_cPPVp). Additionally, we designed an improved variant by further modifying the nPPVp_A:sbmvs:B construct through the fusion of the catalytically inactive C151A mutant version of cPPVp (cPPVp*) with the C-terminus (nPPVp_A:sbmvs:B_cPPVp*) to compensate for the intrinsic affinity of the split protease domains and lower the background activation in the absence of SbMVp activator protease[29]. To evaluate the effectiveness of the PPVp reconstitution, we used a cyclic luciferase reporter with PPVp cleavage site (cycLucPPVs) (Fig. 4f). To evaluate the effectiveness of inhibition and thus determine the background

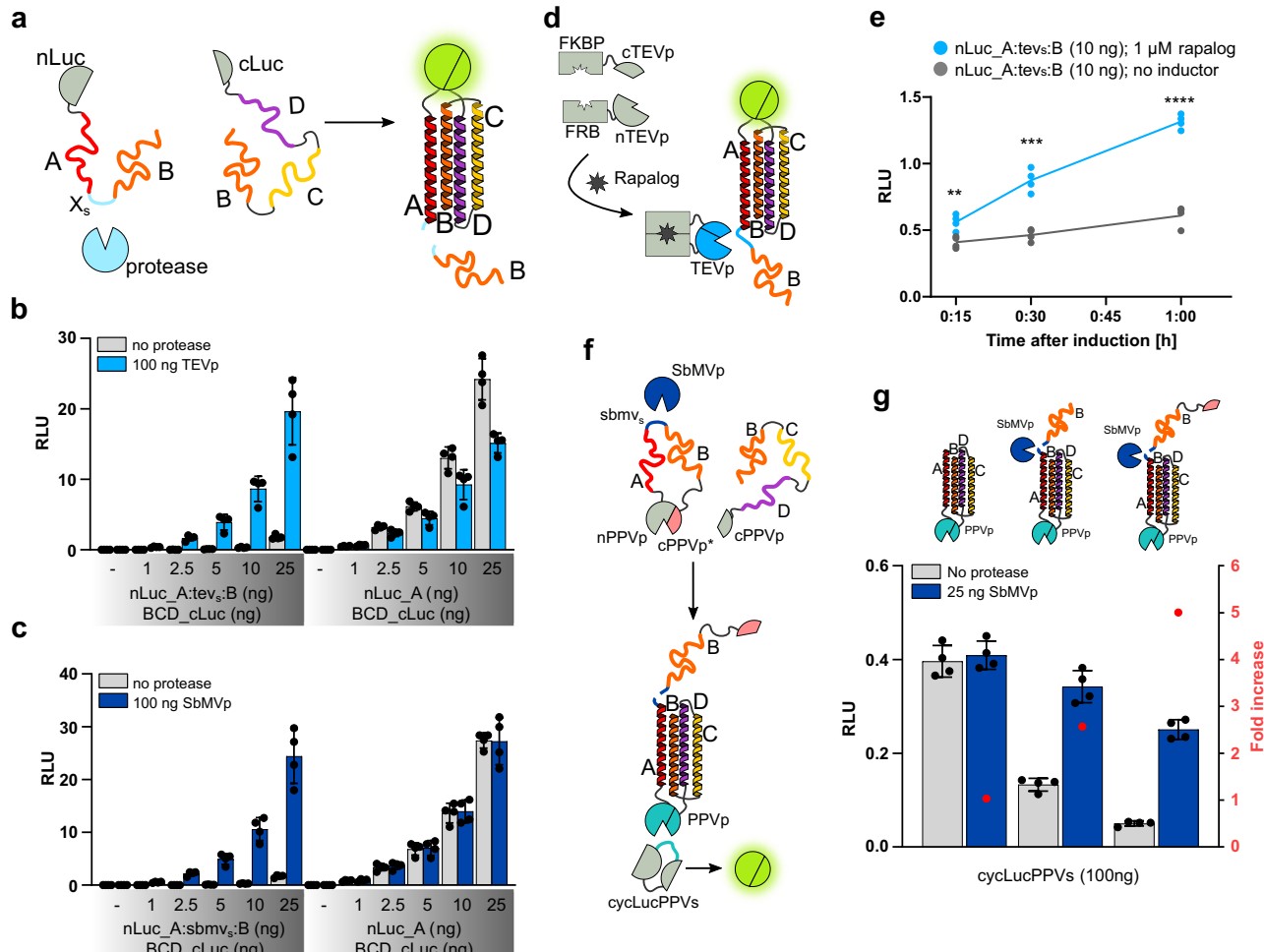

**Fig. 4 | Dimerization of split4HB domains can be regulated by proteolysis and a chemical signal. a** Schematic of inducible dimerization triggered by proteolytic cleavage. The introduction of a protease cleavable linker between peptides A and B creates an inducible dimerization construct nLuc_A'B. Inhibitory peptide B is cleaved off in the presence of an appropriate protease, resulting in hetero-dimerization of nLuc_A and BCD_cLuc and reconstitution of firefly luciferase. **b–c** Titration of TEVp-inducible (**b**) or SbMVp-inducible (**c**) 4HB dimerization in the absence (gray columns) or presence of TEVp (**b**, light blue columns) or SbMVp (**c**, dark blue columns) compared to non-inducible 4HB dimerization. **d** Schematic of exogenously regulated 4HB dimerization by rapalog-inducible splitTEVp. The addition of rapalog leads to FKBP and FRB heterodimerization, resulting in active TEVp, cleaving off the inhibitory peptide B. **e** Kinetics of exogenously regulated 4HB dimerization in the absence (gray dots) and presence (blue dots) of 1 μM rapalog. HEK293T cells were transiently co-transfected with 10 ng of plasmids for split4HB dimerization domains (nLuc_A:tevs:B, BCD_cLuc) and 50 ng of plasmids for the FKBP-cTEVp and FRB-nTEVp constructs. Values are single data points of four

biological replicates and representative of three independent experiments. Two-tailored unpaired *t*-test: 15 min, *p* = 0.0058; 30 min, *p* = 0.000144; 1 h, *p* = <0.0001. **f** Schematic of the designed protease cascade. SbMVp-inducible split4HB dimerization domains are fused with splitPPVp domains. Co-expression of SbMVp results in cleavage of inhibitory peptide B fused with the inactive cPPVp* domain. Enabled 4HB dimerization brings PPVp domains into proximity sufficient for reconstitution. Subsequent cleavage of cyclic firefly luciferase reporter (cycLucPPVs) results in active firefly luciferase. **g** Top: Protease cascade setups. Non-inducible (left), SbMVp-inducible split4HB dimerization without (middle) and with fusion to inactive cPPVp* domain (right). Bottom: Evaluation of protease cascade setups in the absence (gray columns) and presence of SbMVp (blue columns). Red dots indicate a fold increase in signal in the absence and presence of SbMVp. HEK293T cells were transiently transfected with 2.5 ng plasmids encoding protease cascade constructs. Values in (**b**, **c**, and **g**) are the mean of four biological replicates ± (s.d.) and representative of three independent experiments.

reconstitution of splitPPVp, we compared the newly designed constructs with a non-inducible PPVp reconstitution on the split4HB dimerization domains in the absence of SbMVp. We observed 3- and 8-fold decreases in the constructs with inhibitory peptide B and its fusion with cPPVp*, respectively (Fig. 4g), which indicated a successful inhibition of A:BCD interactions with inhibitory peptide B. Comparing luciferase activities in the absence and presence of SbMVp, we detected 2.5- and 5-fold increases for nPPVp_A:sbmvs:B and nPPVp_A:sbmvs:B_cPPVp*, respectively (Fig. 4g). At 25 ng of co-transfected SbMVp plasmid, the SbMVp-inducible splitPPVp cascade reconstitution reached the maximum signal, comparable to the native PPVp (Supplementary Fig. 4b). The same principle could be used for the regulation of 2:2 segmentation modules by positioning

the protease cleavage site between the B and C segments in the BCD construct, which would enable AB:CD pairing upon proteolytic cleavage.

## Applicability of 1:3 segmentation strategy to different designed 4HBs

To test the general applicability of the segmentation strategy to other designed 4HBs, segmentation into 1:3 dimerization modules was tested on three additional designed 4HBs: DHD9 (PDB: 5J73)[26], DHD15[25], and DHD37[25] (Fig. 5). This segmentation was selected since 1:3 dimerization can reveal the contribution of each helical peptide to the 4HB assembly. In DHD9 and DHD15, the experimental analysis demonstrated that there are individual helices that are not required for the

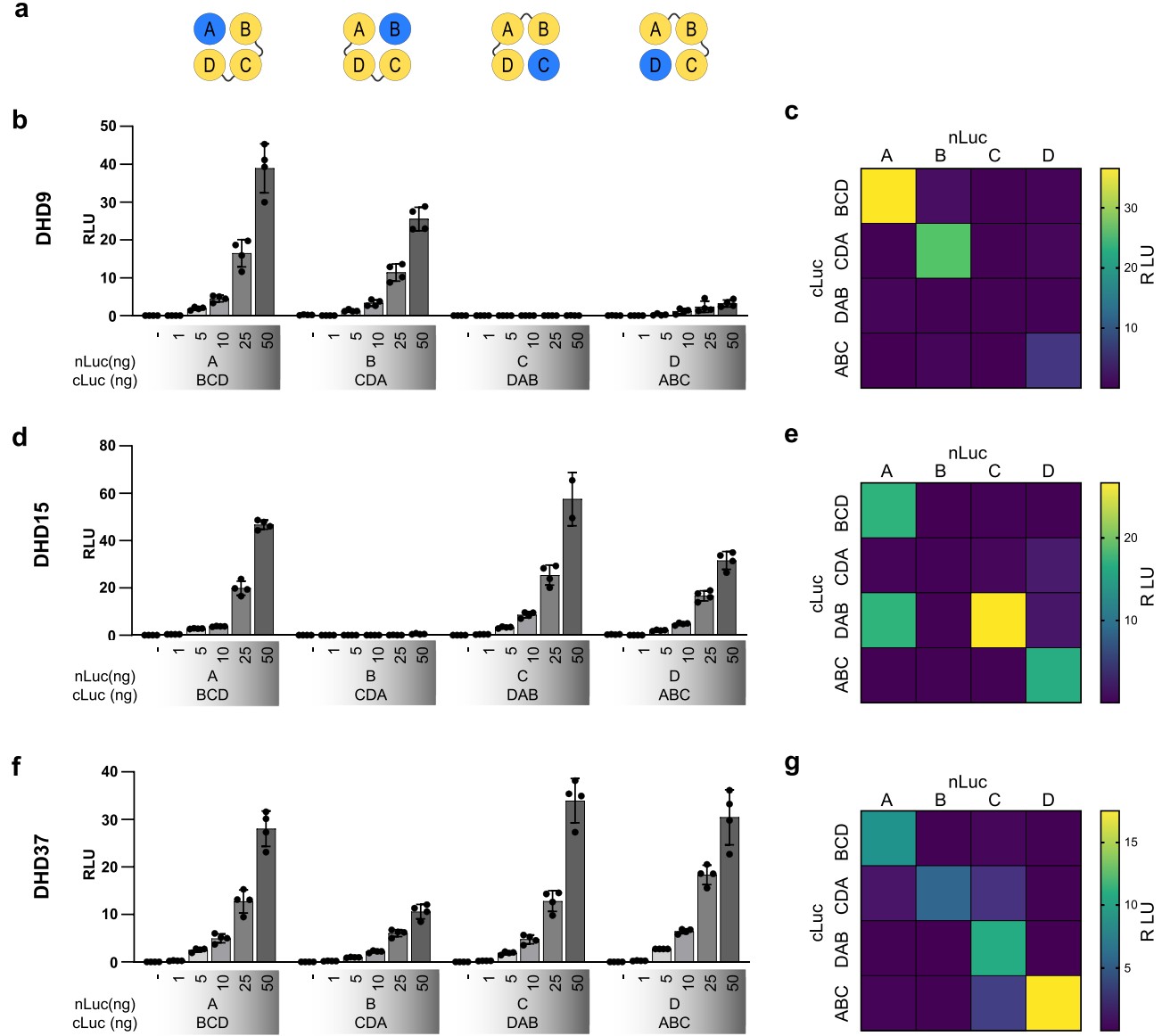

**Fig. 5 | Four-helical bundle splitting strategy is applicable to diverse 4HBs and provides insight into the contribution of each segment to helical bundle formation. a** Schematic representation of the generated 1:3 dimerization modules. **b**, **d**, **f** Titration of DHD9 (**b**), DHD15 (**d**), DHD37 (**f**) 4HB 1:3 dimerization domains. **c**, **e**, **g** Orthogonality of designed DHD9 (**c**), DHD15 (**e**), DHD37 (**g**) 4HB 1:3 dimerization domains. HEK293T cells were transiently transfected with 25 ng plasmids encoding for dimerization constructs. Values in (**b**, **d**, **f**) are the mean of four biological replicates ± (s.d.) and representative of three independent experiments. Values in (**c**, **e**, **g**) are the mean of four biological replicates and representative of three independent experiments.

assembly of a helical bundle (Fig. 5b–e), suggesting that a 3HB lacking a particular helical segment may have stability comparable to that of a 4HB. Nevertheless, we identified a 4HB DHD37 with prominent contributions from all four helices to the 4HB assembly with higher orthogonality, compared to the initially selected DHD13_XAAA 4HB for the investigation of the segmentation strategy (Fig. 5f, g).

Information gathered from this type of segmentation can be valuable for discovering problematic areas in helical bundles, which can escape the rational design evaluation if only the stability of the whole 4HB is considered as the metric. Additionally, we evaluated the predicted helical bundle formation using AlphaFold2 modeling[32,33], which indicated that the 3HB assemblies lacking one helical peptide may also be formed in addition to 4HB. Due to the weak correlation with stability, AlphaFold2 can currently not predict which helical segment has a weak contribution to 4HB stability and if alternative helical bundles, such as 3HB, might have comparable stability (Supplementary Fig. 5). We identified such a problem in the DHD13_XAAA 4HB,

where segmentations with a peptide C as a single peptide in the construct reconstituted the split luciferase in neither 1:3 dimerization nor 2:1:1 trimerization arrangement. In these cases, AlphaFold2 modeling predicted the expected 4HBs as well as the assembly of the 3HB lacking peptide C. However, AlphaFold2 exhibited similar predictions also in cases where a single peptide, e.g., A, was required for di- or trimerization (Supplementary Fig. 5). The strategy of 4HB segmentation into 1:3 dimerization domains thus provides a valuable and generally applicable platform for evaluating the contribution of each peptide to 4HB formation. Furthermore, we can specifically identify which peptide should be targeted for further optimization of the designed structure to stabilize the helical bundle with better-designed interaction specificity.

## Construction and evaluation of trimerization domains
Next, we investigated the trimerization and tetramerization segmentations. In these cases, single peptide modules were fused with the

fluorescent protein mCherry or YFP to increase their stability, as non-structured peptides are prone to rapid degradation in mammalian cells. Compared to the expression of single peptides in mammalian cells, fusion with fluorescent proteins significantly increased the reconstitution of trimerization (Supplementary Fig. 6a) and tetra-merization combinations (Supplementary Fig. 6b). In addition to the stabilization of the unstructured peptide, fusion with fluorescent proteins further demonstrates the functionalization possibility of single-module peptides. This supports the idea of bringing three or four unrelated proteins into close proximity in a desired geometric arrangement and stoichiometry. To test the dynamic range, we evaluated A:B:CD trimerization and A:B:C:D tetramerization in a concentration-dependent manner, where the titration of the "trigger" mCherry_B domain was independent of the nLuc and cLuc constructs. The trimerization surprisingly demonstrated even higher luciferase activity than single-chain 4HB (Supplementary Fig. 6c), most likely due to the increased stability based on mCherry fusion. We further evaluated the influence of mCherry fusion on trimerization reconstitution by titration of "trigger" peptides. In the presence of mCherry_B as a "trigger," peptide luciferase reconstitution reached the plateau at 5 ng; however, the expression of only peptide B did not reach the plateau even at 150 ng of plasmid (Supplementary Fig. 6d).

Next, we assessed all possible trimerization combinations in the split luciferase reporter system. The amount of luciferase reconstitution in the absence and presence of the third "trigger" peptide was evaluated, providing the background and response signals for each trimerization arrangement. As with dimerization domains, we observed different levels of split luciferase reconstitution based on different segmentation and fusion arrangements of luciferase parts and fluorescent proteins (Fig. 6a, Supplementary Fig. 7a, b). The characterization of trimerization domains further supports the identification of weak peptide C contribution to the helical bundle formation since two trimerization arrangements with C peptide as a separate chain showed significantly lower reconstitution of the split luciferase (Fig. 6a, Supplementary Fig. 7a, b). Combinations A:B:CD and A:D:BC, in contrast, demonstrated high signal-to-background ratio.

The tetramerization of four single helix peptides was, on the other hand, also demonstrated but with slightly lower reconstitution efficiency (Supplementary Fig. 6e). We observed a very low background of luciferase activity and successful reconstitution of luciferase when the "trigger" peptide mCherry_B was co-transfected (Fig. 6b). Interestingly, in case of tetramerization arrangement peptide all four peptides, including peptide C, were required for the assembly, which suggests that the covalently linked helices may contribute to the stabilization of a 3HB, which is not favored in a tetramerization arrangement.

### Control of gene expression based on trimerization

With the newly designed trimerization domains, we wanted to assess the possibility of regulating gene expression. To test this principle, we designed a tripartite synthetic activator labeled X:Y:Z with TALE[A] DNA-binding domain fused with one of the trimerization domains (X) and the VP16 activator domain fused with the second trimerization domain (Y), and the third domain serving as the "trigger" of trimerization (Z). Upon expression of the "trigger" peptide, the trimerization domains assembled into a 4HB, which resulted in close proximity to TALE and VP16, enabling transcription of the reporter protein firefly luciferase under a minimal promotor flanked by ten or a single TALE[A]-binding site (Fig. 6c). TALE[A] and VP16 fusion constructs were simultaneously expressed without noticeable interaction and thus with a very low background transcription activation level. The best activation of transcription was achieved with A:CD:B trimerization segmentation (Fig. 6d). This trimerization combination displayed a wide dynamic range, achieving high reporter expression even with 1 ng of the "trigger" peptide plasmid. We also obtained a significant increase in transcription activation with CD:B:A and D:BC:A

trimerization combinations on the reporter with either ten (Fig. 6d) or a single TALE[A]-binding site (Supplementary Fig. 8a). As expected, no activation was observed for DA:C:B. The concentration dependence of the "trigger" peptide B with or without fusion with the fluorescent protein mCherry was also evaluated in a TALE-VP16 system. The lower response amplitude of mCherry_B_NLS compared to B_NLS might be due to a steric hindrance of mCherry proteins clustering at the TALE[A] DNA-binding sites, which may hinder the reconstitution of trimerization domains on the DNA (Supplementary Fig. 8b).

### Doxycycline-inducible trimerization

Another type of inducible activation of oligomerization is transcriptional regulation with the Tet-On 3G system, based on the doxycycline responsive element (Tet-On 3G) under the constitutive promotor and protein of interest under the minimal promotor flanked by the Tet-On 3G DNA-binding site (pTRE3G)[34,35]. The addition of doxycycline allows Tet-On 3G to bind DNA and activate the transcription of a protein of interest (peptide B). We constitutively expressed two trimerization domains (nLuc_A and CD_cLuc) and the doxycycline-inducible production of the third ("trigger") trimerization domain (B) to achieve an "on-demand" trimerization (Fig. 6e). The induction of peptide B transcription with 100 ng/ml doxycycline resulted in a 14-fold increase in the firefly luciferase signal due to trimerization (Fig. 6f). The amount of luciferase activity of inducible trimerization was comparable to both direct trimerization reconstitution of the split luciferase and the transcriptional activation of the firefly luciferase under the pTRE3G promotor in the presence of an inductor (Fig. 6f).

### CAR-T-designed signaling by trimerization

Cancer immunotherapy based on chimeric antigen receptor T (CAR-T) cells is a cell therapy in which the tight control of the regulation of a CAR receptor signaling is desirable[36–40]. CAR-T cells recognize antigens presented on cancer cells and eliminate them. They are designed as a fusion of the CD3ζ activation domain of the T cell receptor or its variations with extracellular antigen-binding single-chain variable fragment (scFv) domains. CAR-T cell technology presents a milestone in cellular therapy; however, antigen escape presents limitations for its wider application[37]. Antigen escape and relapse of cancer tissue after remission can be a consequence of two events: downregulation of the targeted surface antigen or overgrowth of tumor cells that do not express the targeted antigen[37–39]. Different strategies have been proposed for resolving these limitations[37,38], with dualCAR-T systems emerging as powerful tools[36,41–45] aimed to select the target cells expressing at least one of the two antigens of interest. To regulate CAR-T cell response to the antigens present on tumor cells, one can design artificial signal processing pathways by introducing designed protein–protein interactions. This idea was implicated in the remote control of CAR-T cells[46,47] and as a split, universal, and programmable (SUPRA) CAR system[48]. These systems are based on introducing protein–protein interactions in CAR-T cells.

We investigated the segmentation technology of 4HB comprising heterotrimerization for potential therapeutic applications in CAR-T cell technology. We developed a platform for dual CAR-T cell technology designed by the direct fusion of anti-CD19 scFv and anti-CD20 scFv through a transmembrane domain and 4-1BB co-stimulatory domains fused to the A and CD modules of 4HBs, respectively. The direct fusion of peptide B with the CD3ζ signaling domain for the activation of CAR-T cells allowed its recruitment to the complex. We prepared peptide B in various configurations with CD3ζ signaling domain and the 4-1BB co-stimulatory domain (Supplementary Fig. 9a). In this system, we expect peptides A:B:CD to assemble into a 4HB; however, the presence of CD19 or CD20 at the surface of cancer cells should result in CAR clustering and activation of T cells. We tested the systems in Jurkat cells for all B trimerization domain variations independently. The stimulation with Raji cells, which express high levels of CD19 and CD20 ligands, resulted

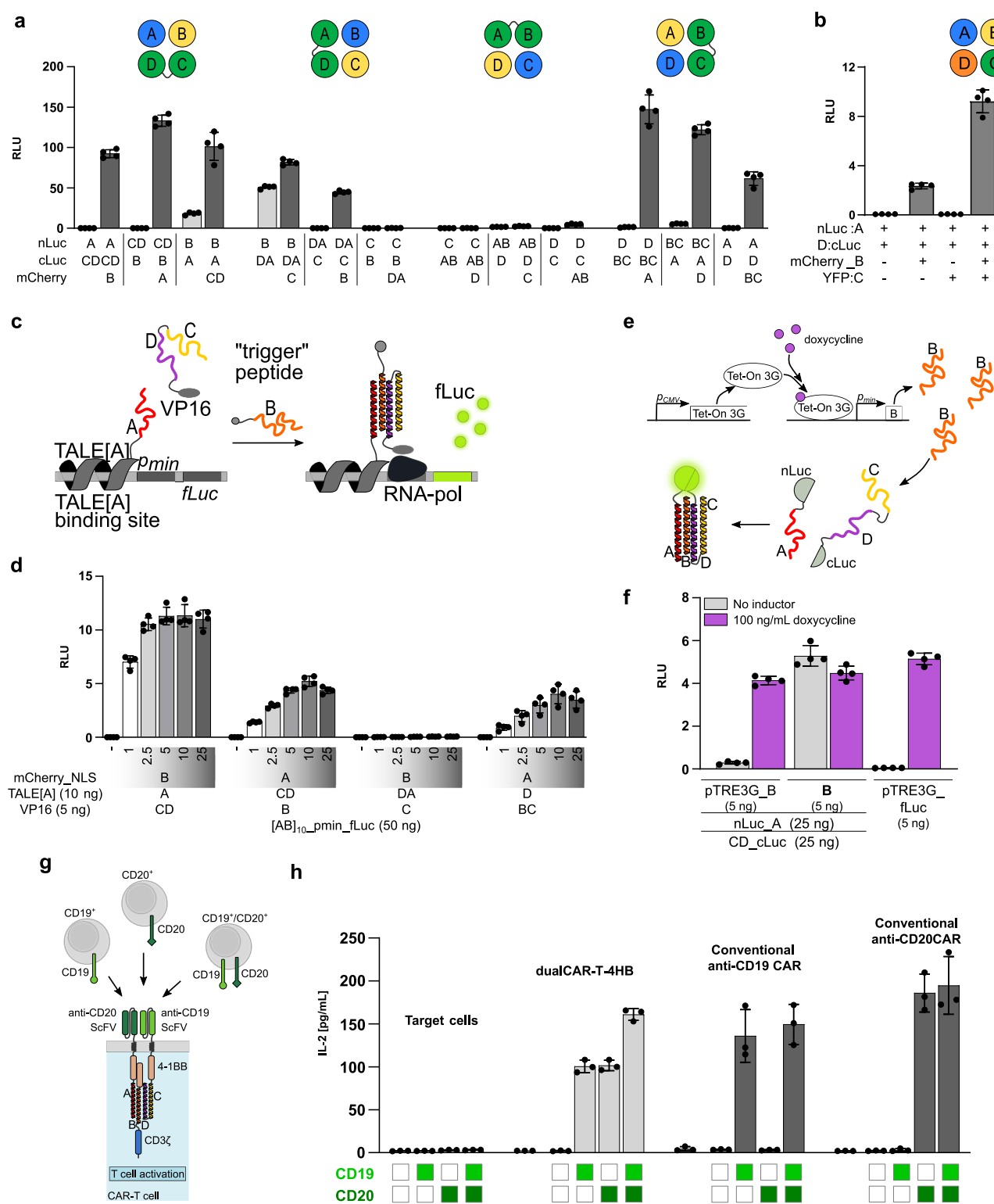

in the highest hIL-2 production in the case of CAR-T-4HB with the trimerization domain variant BIIb (4-1BB_B_CD3ζ) when targeted against CD19 (Supplementary Fig. 9b) or CD20 (Supplementary Fig. 9c) ligands. Furthermore, to combat antigen escape, we created dualCAR-T-4HB from trimerization modules, designed to target either CD19 or CD20 ligands, which could be straightforwardly exchanged for any other combination of scFvs. dualCAR-T-4HB is comprised of a membrane-anchored anti-CD20 scFv fused with intracellular 4-1BB and peptide A, a membrane-anchored anti-CD19 scFv fused with intracellular 4-1BB and CD peptides, and an intracellular BIIb trimerization domain variant (4-

1BB_B_CD3ζ) (Fig. 6g, Supplementary Fig. 10a). We observed comparable expression levels of second-generation anti-CD19 CAR, anti-CD20 CAR, and dualCAR-T-4HB (Supplementary Fig. 10b, c). Upon stimulation of the CAR construct expressing Jurkat cells with Raji cells, the designed dualCAR-T-4HB induced hIL-2 production comparable to the conventional second-generation anti-CD19 CAR or anti-CD20 CAR and order of magnitude higher than hIL-2 levels produced by tandem CARs using the same anti-CD19 and anti-CD20 scFvs connected through a flexible linker (G4S)5[49] (19/20-tanCAR and 20/19-tanCAR) (Supplementary Fig. 11). dualCAR-T-4HB presents advancement toward a universal modular

**Fig. 6 | Design, characterization, and biological application of 4HB trimerization and tetramerization domains. a** Comparison between 4HB trimerization background signal−only nLuc and cLuc fusion constructs (light gray columns) and response signals−co-transfection with "trigger" peptide (dark gray columns). **b** Reconstruction efficiency of tetramerization 4HB domains. Comparison between background signal with nLuc_A and D_cLuc and response signals−co-transfection with YFP_C and mCherry_B. HEK293T cells in (**a**) and (**b**) were transiently transfected with 25 ng of constructs. **c** Schematic of transcription activation using trimerization of split4HB domains. A DNA-binding protein TALE[A] is covalently linked with the first trimerization domain by the GS linker and NLS signal. The second trimerization domain is fused with the VP16 transcriptional activator through the GS linker and NLS. A third trimerization domain acts as a "trigger" peptide in fusion with mCherry through a flexible GS linker and NLS. **d** Determination of trimerization efficiency from different split4HB trimerization domains by TALE[A] and VP16 proximity-related transcriptional activation of firefly luciferase. **e** Schematic of doxycycline-inducible 4HB trimerization. Tet-On 3G protein expression is under a constitutive promotor. Peptide B acts as a

trimerization "trigger" and is under an inducible promotor that comprises a Tet-On 3G-binding site and a minimal promotor. **f** Comparison of non-inducible trimerization (middle) with doxycycline-inducible split4HB trimerization (left) and doxycycline-inducible firefly luciferase (right) in the absence (gray columns) and presence of doxycycline (purple columns). **g** Schematic of designed dualCAR-T-4HB based on 4HB trimerization domains, comprising anti-CD19 scFv fused with CD peptides through 4-1BB co-stimulatory domain; anti-CD20 scFv fused with peptide A through 4-1BB co-stimulatory domain; CD3ζ activator domain fused with peptide B flanked by 4-1BB co-stimulatory domain. Co-cultivation with single-positive CD19⁺, CD20⁺, or double-positive CD19⁺/CD20⁺ cells leads to the activation of dualCAR-T-4HB cells and hIL-2 production. **h** Analysis of hIL-2 production of dualCAR-T-4HB, CD19-CARwt, and CD20-CARwt introduced into Jurkat cells after stimulation with double-negative K562, single-positive K562-CD19⁺ or K562-CD20⁺, or double-positive K562-CD19⁺/CD20⁺ cells. Values in (**a**, **b**, **d**, **f**) are the mean of four biological replicates ± (s.d.), and values in (**h**) are the mean of three biological replicates ± (s.d.) and representative of three independent experiments.

platform for targeting two antigens without the need for scFv optimization. To evaluate the specific ligand targeting ability of dualCAR-T-4HB, single-positive K562-CD19⁺ and K562-CD20⁺ and double-positive K562-CD19⁺/CD20⁺ cell lines were prepared (Supplementary Fig. 12a). The expression of CD19 and CD20 ligands was determined by flow cytometry (Supplementary Fig. 12b). Compared to the conventional second-generation anti-CD19 CAR and anti-CD20 CAR, which produce hIL-2 only in the presence of the specific targeted antigen, the stimulation of dualCAR-T-4HB transfected Jurkat cells with target cells K562, K562-CD19⁺, K562-CD20⁺, and K562-CD19⁺/CD20⁺ resulted in hIL-2 production in the presence of CD19, CD20, or both antigens, but not when neither of the ligands was present on the target cell surface (Fig. 6h). Such a response is expected for the OR logic function, which is in agreement with our hypothesis. Noncovalent interaction between the complex receptor segments facilitates the mobility of the recognition complex. These results corroborate the feasibility of the segmented 4HB to facilitate the formation of a heterotrimeric complex that can activate CAR-T cells by either CD19 or CD20 antigen to counter antigen escape in cancer cell therapies. This system is particularly suitable for testing different combinations of sensing and signaling modules.

## Discussion

Here, we present a combinatorial segmentation strategy of helical bundles on the example of 4HB for use in engineering protein interactions for synthetically designed pathways and demonstrate the regulation of mammalian cells, although it should work equally well also in other cell types and in vitro. Dimeric oligomerization domains have been previously used from a single-chain 4HB; however, additional dimerization, trimerization, and tetramerization segmentation strategies investigated here could be applied to other designed or natural 4HBs. Different dimerization and trimerization domain combinations resulted in an increased number of orthogonal combinations and new regulation modalities. This presents the possibility for fine-tuning the desired cellular response by selecting the appropriate oligomerization domains. With further advances in protein design modeling, additional variants of 4HBs will be generated where we can expect to see comparable contributions of all included helices. Apparently, the formation of a highly stable 4HB concealed the weaker contribution from peptide C with the rest of the peptides in the 4HB, which has also not been apparent from the analysis of the 3D structure. This might be because the helices A, B, and D may form a 3HB with slightly weaker or comparable stability to the 4HB based on the rearrangement of their geometry. The prediction of the assembly composed of A, B, and D peptides was plausible based on AlphaFold2 modeling, which also suggested the correct orientation; however, this has to be experimentally verified, as AlphaFold2 is currently not able to quantitatively evaluate the stability of each fold and the effect of

point mutations. Reorientation and even variable stoichiometric promiscuity of coiled-coil peptides have been however demonstrated before; for example, few mutations are sufficient to transform a coiled-coil dimer into a trimer or tetramer[50] and different orientations and oligomerization states of the CCs may coexist. Nevertheless, we demonstrated the functionalization of a 4HB with the fusion of split luciferase domains and fluorescent proteins with the trimerization and tetramerization modules. The fusion of protein domains with peptides retained the reconstitution of the split4HB and even stabilized single peptides, most likely protecting them from degradation. An important result of our research is the characterization of orthogonal dimerization domains and the possibility of simultaneous use of multiple segmentations without cross-talk. Different segmentations of a single 4HB decrease the need for designing additional sets of helical bundles. This represents a useful tool for diverse applications in synthetic biology while bypassing the paucity of oligomerization domains.

To expand the versatility of the designed oligomerization domains, we implemented the investigated segmentations of 4HBs for various biological processes as synthetic biology tools: transcriptional regulators, protease-regulated dimerization modules and protease cascade arrangement, chemically inducible trimerization modules and dual CAR-T modules for engineering different combinations for arming T cells against cancer cells. Conventional engineering of tandem CARs for targeting multiple surface-exposed antigens often requires optimization for each new scFv separately, combining the order of scFvs and their domains to avoid the formation of an artificial scFv and linkers connecting the two scFvs[51]. To overcome these limitations, we present the dualCAR-T-4HB system demonstrated on targeting CD19 and CD20 in an OR function-like fashion that could be readily adapted to diverse combinations of tandem scFv domains and intracellular signaling domains to test different combinations of co-activation domains. The dualCAR-T-4HB system is modular and avoids problems with the folding of scFv segments and their aggregation, which may be an issue for tandem recognition domains. Compared to the Co-LOCKR system, the dualCAR-T-4HB presents only humanized scFvs at the cell membrane, which have been widely applied and tested in vivo in CAR-T cell therapies. However, the immune response to constructs comprising segmented 4HBs needs to be tested. Chimeric receptors could be further developed to include a regulatory element such as inducible 4HB assembly (e.g., Fig. 6f). Different segmentation strategies may also be applied to larger helical bundles for introducing a more complex recognition of the desired combinations of target antigens within a cell or biological complex. Trimerization and tetramerization domains additionally facilitate the introduction of different stoichiometries, co-localization of multiple biological molecules, and increased target selectivity by multiple recognition domains.

## Methods

### Plasmid construction

The coding DNA sequences for DHD13_XAAA, DHD9, DHD15, and DHD37 were codon-optimized for expression in human cells and synthesized by Genewiz, Leipzig, Germany GmbH, Twist Bioscience HQ, South San Francisco, California, USA, or Integrated DNA Technologies, Inc., Coralville, Iowa, USA. PCR amplification was performed using repliQa HiFi ToughMix® (Quantabio, Beverly, MA, USA). Plasmids were constructed using the standard procedures of molecular cloning or Gibson assembly[52]. The amino acid sequences of all constructs are provided in Supplementary Data 1.

### AlphaFold2 modeling

Protein models were built using publicly available scripts and modeling algorithms[32,33].

### Chemical inducers

A/C Heterodimerizer (rapalog, AP21967; Clontech Laboratories, Inc., part of Takara Bio USA, Inc.) was dissolved in dimethyl sulfoxide (Sigma-Aldrich) at 1 mM concentration. Before stimulation, a stock solution of rapalog was diluted in Dulbecco's Modified Eagle Medium (DMEM) medium (Invitrogen) to the final 1 μM concentration and added to each well in 96-well plates. Doxycycline (Sigma-Aldrich) was dissolved in MQ at 5 mg/ml concentration. Prior to the experiment, doxycycline was further diluted in DMEM to the final concentrations used for the stimulation.

### Cell cultures

The human embryonic kidney (HEK) 293T cell line was cultured in DMEM, GlutaMAX™ Supplement (DMEM medium, Invitrogen) supplemented with 10% fetal bovine serum (FBS; BioWhittaker, Walkersville, MD, USA), and the Raji, Jurkat, and K562 cell lines were cultured in the Roswell Park Memorial Institute 1640 Medium, GlutaMAX™ Supplement (RPMI 1640 medium, Invitrogen) supplemented with 10% FBS. The cell lines were maintained at 37 °C in a 5% $CO_2$ environment. Cell lines were obtained from the ATCC culture collection.

### Transfection

HEK293T cells were washed with phosphate-buffered saline (PBS) and detached from the surface using Trypsin-EDTA solution (Sigma-Aldrich; T3924). The cell concentration was measured using Countess™ Cell Counting Chamber Slides or EVE™ Cell Counting Slides kits with Trypan blue as an indicator of live cells and measured on the Countess™ 3 Automated Cell Counter (Invitrogen™). The cells were seeded in white 96-well and 24-well plates (CoStar, Corning) at $2 \times 10^4$ live cells per well and $1 \times 10^5$ live cells per well, respectively. At a confluence of 50–70%, the cells were transfected with a mixture of DNA and PEI (batch-optimized 3–6 μl/500 ng DNA). The PEI stock concentration 0.324 mg/ml, pH 7.5, was diluted in 150 mM NaCl and mixed at a 1:1 ratio with the appropriate DNA, also diluted in 150 mM NaCl. This was incubated at room temperature for 15 min and added to the cell media in plates. To normalize the reporter values to transfection efficiency, 5 ng of constitutively expressed control plasmid phRL-TK (Renilla luciferase encoding plasmid) was used for the dual luciferase assay, and 5 ng of constitutively expressed control plasmid for fluorescent protein iRFP was used for flow cytometry.

### Luciferase assay

HEK293T cells were harvested at the indicated time points after transfection and/or stimulation and lysed with 30 μl of 1× Passive Lysis buffer (Promega). Firefly luciferase and Renilla luciferase expression were measured using the dual luciferase assay (Promega) on an Orion II, Mithras LB 940, or Centro LB 963 microplate reader (Berthold Technologies) using Simplicity 4.2 software. Relative luciferase units (RLUs) were calculated by normalizing the firefly luciferase activity to the constitutive Renilla luciferase activity determined within the same sample.

### Flow cytometry

The transfected HEK293T cells were maintained at 37 °C in a 5% $CO_2$ environment. The medium with transfection elements was changed for fresh DMEM supplemented with 10% FBS at 24 h post-transfection. At 48 h post-transfection, the cells were harvested with Trypsin-EDTA solution and washed twice with PBS supplemented with 10% FBS. The same solution was used for the analysis of samples on the Cytek™ Aurora Flow Cytometry System (Cytek® Biosciences). For flow cytometry controls and unmixing, we used HEK293T cells transfected with an empty vector (pcDNA3) as negative control and constitutive expression vectors encoding for mCitrine, tagBFP, and iRFP as single-strain controls.

### Electroporation

CAR-T effector cells were generated from Jurkat cells by electroporation with a mixture of DNAs using the Neon™ transfection system (Invitrogen™), according to the manufacturer's instructions. Electroporated cells were incubated for 24 h. CAR-T target cells expressing specific ligands—K562-CD19⁺, CD20⁺, and CD19⁺/CD20⁺ cells—were generated from parental K562 cells through electroporation of pEF1α_CD19-mCitrine, pEF1α_CD20-tagBFP, or pEF1α_CD19-mCitrine_pEF1α_CD20-tagBFP plasmid constructs with the Neon transfection system.

### Quantification of expressed proteins in CAR-T system

The expression of CD19 and CD20 on the plasma membrane was analyzed with flow cytometry as the detection of fused fluorescent proteins mCitrine and tagBFP on Cytek™ Aurora Flow Cytometry System (Cytek® Biosciences).

The expression of CAR constructs on the plasma membrane was analyzed with flow cytometry and Western blot by detection of genetically fused N-terminal Myc-tag. For flow cytometry experiments, $5 \times 10^5$ cells were collected 48 h post electroporation and centrifuged for 5 min at 250 g. Pellet was washed with 1 ml PBS supplemented with 10% FBS and centrifuged for 5 min at 250 g. Pellet was resuspended in 80 μl FcR blocking reagent for 10 min at 4 °C. Antibodies against Myc-tag labeled with Alexa Fluor 647 was added in the ratio 1:100 for 30 min at 4 °C. After incubation, cells were washed with the addition of 1.8 ml PBS supplemented with 10% FBS and centrifuged for 5 min at 250 g. Pellet was resuspended in 300 μl PBS supplemented with 10% FBS. Fluorescence was detected with Cytek Aurora spectral flow cytometer using SpectroFlo 10.8.1 software. The rest of the cells were used for Western blot. Cells were collected and centrifuged for 5 min at 250 g. Pellet was washed with 1 ml PBS and centrifuged for 5 min at 250 g. Pellet was resuspended in 100 μl Ripa Lysis buffer supplemented with protease inhibitor cocktail and incubated for 20 min at 4 °C. After incubation, cells were centrifuged for 15 min at 1700 g. Total protein concentration in the supernatant was measured by the Bicistronic acid method. Samples were complimented with denaturant SDS with reducing agent and incubated for 15 min at 95 °C. The total amount of proteins of 10 μg was loaded into Any kD™ Mini PROTEAN® TGX™ Precast Protein Gels with PageRuler™ Plus Prestained Protein Ladder (Thermo Scientific™) for control. SDS-PAGE was run under denaturizing conditions at 200 V for 35 min. Proteins were transferred to 0.45-μm nitrocellulose membrane at 350 mA for 1.5 h. iBind (Thermo Scientific™) was used for antibody incubation. Myc-tagged CAR constructs were specifically detected with primary antibodies Rabbit anti-Myc-tag at 1:2000 and secondary antibodies Goat anti-rabbit-HRP at 1:3000 ratios. For loading control, we detected Hsp70 protein using primary antibodies Mouse anti-Hsp70 at 1:1000 and secondary antibodies Goat anti-mouse-HRP at 1:3000 ratios. Detection of HRP was achieved by incubation of the membrane with SuperSignal™ West

Femto Maximum Sensitivity Substrate (Thermo Scientific™), the loading control membrane part for 5 min, and the CAR for 10 min. The membrane was imaged with G:Box Chemi XT 4 Chemiluminescence and Fluorescence Imaging System (Syngene). Details about the antibodies are as follows:

Anti-c-Myc antibody produced in rabbit (Sigma-Aldrich®, C3956); Anti-Hsp70 antibody [N27F3-4] (Abcam, ab47454)-Mouse monoclonal [N27F3-4] to Hsp70; Goat Anti-Rabbit IgG H&L (HRP) (Abcam, ab6721); Peroxidase AffiniPure Goat Anti-Mouse IgG (H+L) (© Jackson ImmunoResearch Europe Ltd., 115-035-003); Alexa Fluor® 647 anti-c-Myc Antibody (© BioLegend, Inc., 626810)

### Cytokine production quantification
CAR-T effector cells were seeded at $1 \times 10^5$ cells per well in a 96-well plate and co-incubated with Raji as the target cells at an E:T ratio of 10:1 or with K562, K562-CD19$^+$, K562-CD20$^+$, or K562-CD19$^+$/CD20$^+$ as the target cells at an E:T ratio of 5:1 for 48 h. Human IL-2 cytokine concentration in the culture supernatant was detected using an IL-2 uncoated ELISA Kit according to the manufacturer's instructions (Thermo Fisher Scientific, Waltham, MA, USA). Washing between the incubation steps was performed using a HydroSpeed™ plate washer (Tecan). A multiplate reader SinergyMx (BioTek, Winooski, VT, USA) was used to measure endpoint absorbance.

### Software and statistics
FlowJo_v10.8.1 (https://www.flowjo.com/solutions/flowjo) was used to analyze the data obtained from the flow cytometry experiments. Graphs and statistical analyses were prepared with GraphPad Prism.8.4.3 (http://www.graphpad.com/). The values represent the means of at least three experimental replicates (transfections of cell cultures in individual wells) ± standard deviation (s.d.) and are representative of at least two independent experiments. Schemes were prepared with Inkscape™ 1.0.2.0, Brooklyn, NY (https://inkscape.org/).

### Reporting summary
Further information on research design is available in the Nature Portfolio Reporting Summary linked to this article.

## Data availability
The authors declare that the data generated in this study and supporting the findings of this study are available in the paper and its Supplementary Information files. Raw data are available from the corresponding author upon reasonable request. The original 4HB design structures for some 4HBs used in this study are available in the Protein Data Bank (PDB) database under the following accession numbers: DHD13_XAAA (PDB:6DMP); DHD9 (PDB: 5J7). Source data are provided with this paper.

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

## Acknowledgements

This research was supported by grants from the Slovenian Research Agency (research core funding no. P4-0176). We thank Zibo Chen and Ajasja Ljubetič for providing the sequences of DHD13_XAAA four-helical bundle and for valuable advice. We are thankful to Anja Trupej for cloning a few of the plasmid constructs; Robert Bremšak, Darija Oven, Irena Škraba, Bojana Stevović, Anja Perčič, Tina Strmljan, Klementina Podgoršek, and Sanjin Lulić for technical assistance.

## Author contributions

E.M. and B.M. prepared plasmid constructs, performed and analyzed experiments. E.M. and R.J. designed experiments, analyzed experimental data, and wrote the manuscript. R.J. conceived and supervised the study. All authors discussed and commented on the manuscript.

## Competing interests

The authors declare no competing interests.
