## [Peer Review File · Nature Communications]

Reviewers' Comments:

Reviewer #1:

Remarks to the Author:

The paper by Merljak and colleagues describes the efforts to create components based on protein oligomerization. The general strategy was to create such components by a strategy dubbed by the authors of segmentation, which is generally described as design by split constructs. Overall, the paper is clear and the science/data is of high quality. And the authors show that these components have a realm of applications in synthetic biology which would render them very useful. It is my impression that the claims about the utility in CAR-T cells are exaggerated and I make specific points about this below.

Comments

-“Researches have investigated dimerization domains, which spontaneously interact with their designed partner, bringing the two fused proteins into close proximity.”
Rephrase due to awkward English construction

-“determining the rules for their rational design,13 and a de novo design of synthetic helical bundles”
Rephrase due to awkward English construction

-“ However, due to their small size and α -helical design restrictions, orthogonal set of 4HBs have a finite size”

Rephrase due to awkward English construction

-Data shown in figure 3b in terms of orthogonality is far from optimal and there seems to be significant cross talk between different components. One question that this data raises -

-Data shown in figure 3c is not sufficiently well explained so that a non-expert reader can follow the data – for instance I can't tell what the colors of the plot look like

-“ DualCAR-T-4HB presents advancement toward an universal modular platform for targeting two antigens without the need for scFv optimization.” - its unclear to me how the 4HB split constructs achieve this specially in the light that many dual CARs have been constructed without the split modules

-“ Conventional engineering of tandem CARs for targeting multiple surface-exposed antigens often requires optimization for each new scFv separately, combining the order of scFvs and their domains to avoid the formation of an artificial scFv and linkers connecting the two scFvs.” – despite the citation the authors have added many dual specificity CARS have been constructed just by using different single scFvs

- “Compared to the Co-LOCKR system, the DualCAR-T-4HB presents only humanized scFvs at the cell membrane, which are non-immunogenic, are well-characterized and have been widely applied to CAR-T therapies. Chimeric receptors can be further developed for regulation by inducible oligomerization triggered by chemically inducible 4HB (e.g., Fig. 6f) for licensing CAR-T cell activation.” – these claims are in some way overblown, 4HB remains a synthetic construct the issues with immunogenicity will still be there

-The authors mention the possibility of such constructs to be useful in vitro – did the authors characterized these split constructs in vitro, if yes it would be great to have a glimpse of affinities and stabilities. If they tried and the constructs did not work – this would be valuable information that should be included in the paper and would not detract from the value o the overall story.

Reviewer #2:

Remarks to the Author:

This paper by Merljak et al describes the use 4 helical bundle (4HB) proteins as tools to hetero-multimerizing proteins of interest. The authors have demonstrated orthogonal dimerization, regulatability by small molecules and protease activity, control of gene expression, CAR activation. The use of 4HB in synbio circuit engineering is novel. Although there seems to be a large variation in the output expression level, the data seems robust. Overall, they have demonstrated the versatility of the system and some interesting potential applications. After addressing the following comments, I think this work is suitable for publication in Nature Communications.

Protein expression: It is unclear how well the proteins are expressed when fused to 4HB, especially when used in the CAR.

Unique capability: While the multimerization of 4HB is unique, there is no description of how their unique multimerization capability can lead to novel functions. Much of the applications shown in this work can be achieved with other more well-established dimerization domains, such as leucine zipper or PDZ domains. More descriptions that can delineate the novel function that can be created by using 4HB would be very helpful.

Figure 4e time course is interesting, showing rapid activity. It would be interesting to know if the experiment would carry out longer, would the activity plateau.

Some of the RLU ranges from 0.5 (e.g., Figure 4g) to 150 (e.g., Figure 6a). Do they really have that much difference in RLU, or is it an artifact between different experiments? Since Luc is very sensitive, even a small level of expression can be detected. But it will not be very useful if the proteins are not expressed well. However, I do acknowledge that the dualCAR-T 4HB seems to work well.

Dear Editor,

We would like to submit the revised version of the manuscript, entitled “**Segmentation strategy of *de novo* designed four helical bundles expands protein oligomerization modalities for cell regulation**».

We would like to thank the editor and reviewers for their efforts and their thoughtful comments with clear and concise suggestions for the improvements of the presented manuscript. Thank you for your patience while we prepared the revised manuscript.

Here is our point-by-point response to the comments:

Reviewer #1 (Remarks to the Author):

The paper by Merljak and colleagues describes the efforts to create components based on protein oligomerization. The general strategy was to create such components by a strategy dubbed by the authors of segmentation, which is generally described as design by split constructs. Overall, the paper is clear and the science/data is of high quality. And the authors show that these components have a realm of applications in synthetic biology that would render them very useful. It is my impression that the claims about the utility of CAR-T cells are exaggerated and I make specific points about this below.

We thank the referee for her/his careful and insightful review of our manuscript. We address the issue raised by the referee below.

Comments

-“Researches have investigated dimerization domains, which spontaneously interact with their designed partner, bringing the two fused proteins into close proximity.”

Rephrase due to awkward English construction

Response: We have rewritten the sentence to: “Researches have investigated dimerization domains, which spontaneously interact with their designed partner. When genetically fused to non-interacting proteins, dimerization domains serve as a tool for bringing the two fused proteins into close proximity.”

-“determining the rules for their rational design,¹³ and a *de novo* design of synthetic helical bundles”
Rephrase due to awkward English construction

Response: We have rewritten the sentence to: “The general 4HB structure has been extensively studied,⁸⁻¹⁰ which facilitates the re-design of naturally occurring 4HBs^{11,12} and determining the rules for *de novo* design of synthetic 4HBs.¹³⁻¹⁵”

-“ However, due to their small size and α -helical design restrictions, orthogonal set of 4HBs have a finite size”
Rephrase due to awkward English construction

Response: We have rewritten the sentence to: “However, sets of orthogonal 4HBs are likely to have a limited number of members, due to design restrictions within the layers of 4HBs.²⁵”

Although we used a professional editing service for the final edit of the manuscript, some formulations were clearly not optimal, which we have now tried to revise for the above-mentioned sentences.

-Data shown in figure 3b in terms of orthogonality is far from optimal and there seems to be significant cross talk between different components. One question that this data raises -

Response: In biological systems orthogonality is rarely completely black and white, nevertheless, data demonstrate several pairs that have excellent orthogonality and clearly increase the size of the maximal orthogonal set in comparison to previous segmentations. For some cases, we can provide some explanation for lack of orthogonality. For instance, peptide pair B and DA had a tendency to interact, confirmed also with other experimental data (see trimerization modules on Fig. 6a), which might be due to the lower contribution of peptide C to the formation of 4HB; this is discussed in the results and discussion section. We do not know the cause of all cases of decreased orthogonality, but one possible explanation might be formation of higher-order assemblies with sufficient stability. However, problems with orthogonality of 1:3 dimerizations have also been observed in other experimentally evaluated 4HBs (DHD9, DHD15 and DHD37; see Fig. 5). Taken together, this experimental data support our conclusion of the value of segmentation strategy to the evaluation of the *de novo* 4HB designs, and discovering problematic peptides through their contribution to the assembly of the designed 4HBs.

-Data shown in figure 3c is not sufficiently well explained so that a non-expert reader can follow the data – for instance I can’t tell what the colors of the plot look like

Response: We have amended the explanation of the data in Fig 3c, which should be now easier to comprehend.

Additional explanation included in the figure description:

“Front to back: Reporter plasmids only (gray line); reporter plasmids, TALE[F]-A and TALE[E]-D, (blue line); reporter plasmids, TALE[F]-A, TALE[E]-D and BCD:VP16 activator domain (orange line; expecting mCitrine expression); reporter plasmids, TALE[F]-A, TALE[E]-D and ABC:VP16 activator domain (green line; expecting tagBFP expression); reporter plasmids, TALE[F]-A, TALE[E]-D, BCD:VP16 and ABC:VP16 (brown line; expecting mCitrine and tagBFP expression).”

The next two comments closely related, therefore we reply to them together.

-“ DualCAR-T-4HB presents advancement toward an universal modular platform for targeting two antigens without the need for scFv optimization.” - its unclear to me how the 4HB split constructs achieve this specially in the light that many dual CARs have been constructed without the split modules

-“ Conventional engineering of tandem CARs for targeting multiple surface-exposed antigens often requires optimization for each new scFv separately, combining the order of scFvs and their domains to avoid the formation of an artificial scFv and linkers connecting the two scFvs.” – despite the citation the authors have added many dual specificity CARS have been constructed just by using different single scFvs

Response : In our hands, tandem CARs without optimization demonstrated low activity (See **Supplementary Fig. 11**) and seems to require additional optimization, which is often not described in the papers. The modular nature of segmented 4HB removes at least part of those constraints including the need for linker optimization and should make it useful for the analysis of different combinations of targeting domains of CARs, which may be, if desired, later optimized as a fusion protein.

/note: **Supplementary Fig. 11: Comparison of dualCAR-T-4HB to conventional and tandem CAR constructs.** is numbered by revised manuscript numbering of supplemental figures/

- “Compared to the Co-LOCKR system, the DualCAR-T-4HB presents only humanized scFvs at the cell membrane, which are non-immunogenic, are well-characterized and have been widely applied to CAR-T therapies. Chimeric receptors can be further developed for regulation by inducible oligomerization triggered by chemically inducible 4HB (e.g., Fig. 6f) for licensing CAR-T cell activation.” – these claims are in some way overblown, 4HB remains a synthetic construct the issues with immunogenicity will still be there

Response: We agree that the potential immunogenicity of designed 4HBs may be an issue that might need to be evaluated in the future. On the other hand, de novo designed regulators, such as 4HBs,

permit modifications of surface-exposed residues as B cell epitopes that could be optimized for reduced immunogenicity, such as introduction of polar and charged residues .

We have modified the text as suggested:

“Compared to the Co-LOCKR system, the DualCAR-T-4HB presents only humanized scFvs at the cell membrane, which have been widely applied and tested in vivo in CAR-T cell therapies. However, the immune response to constructs comprising segmented 4HBs needs to be tested. Chimeric receptors could be further developed to include regulatory element such as e.g. inducible 4HB assembly (e.g., Fig. 6f).”

-The authors mention the possibility of such constructs being useful in vitro – did the authors characterized these split constructs in vitro, if yes it would be great to have a glimpse of affinities and stabilities. If they tried and the constructs did not work – this would be valuable information that should be included in the paper and would not detract from the value o the overall story.

Response: We have not analyzed the split constructs *in vitro* as the scope of this manuscript was to focus on the segmented 4HBs as a synthetic biology tool for mammalian cells.

Reviewer #2 (Remarks to the Author):

This paper by Merljak et al describes the use 4 helical bundle (4HB) proteins as tools to hetero-multimerizing proteins of interest. The authors have demonstrated orthogonal dimerization, regulatability by small molecules and protease activity, control of gene expression, CAR activation. The use of 4HB in synbio circuit engineering is novel. Although there seems to be a large variation in the output expression level, the data seems robust. Overall, they have demonstrated the versatility of the system and some interesting potential applications. After addressing the following comments, I think this work is suitable for publication in Nature Communications.

We thank the reviewer for her/his careful and insightful review of our manuscript and kind opinion. We address the issues raised by the referee bellow.

Protein expression: It is unclear how well the proteins are expressed when fused to 4HB, especially when used in the CAR.

Response: We have performed additional experiments to determine the effect peptide fusion on protein expression level using flow cytometry and Western blot, to determine the amount of surface-expressed CARs. Expression of construct comprising 4HB (anti-CD20 scFv fused with intracellular 4-1BB and peptide A) was compared to the conventional 2nd generation anti-CD19 CAR or anti-CD20 CAR by detection of Myc-tag, genetically fused to the N-terminus of the proteins of interest. Antibodies against Myc tag, fluorescently labeled or unlabeled and targeted with HRP-labeled secondary antibody were used for flow cytometry and Western blot, respectively. We detected 20-22% of positive cells in all three examined CAR constructs with flow cytometry. Comparable amounts of expressed CAR constructs were also confirmed with Western blot, suggesting that fusion with 4HB segment does not affect protein production.

Those results were included in an additional Supplementary Figure and cited it accordingly in the main text. Furthermore, we expanded and rearranged the method section, to describe the experimental procedures used, accordingly.

Additional text was included in the main text of the manuscript:

“We observed comparable expression levels of 2nd-generation anti-CD19 CAR, anti-CD20 CAR and dualCAR-T-4HB (Supplementary Fig. 10).”

Changes in the method section:

“Quantification of expressed proteins in CAR-T system

The expression of CD19 and CD20 on the plasma membrane was analyzed with flow cytometry as the detection of fused fluorescent proteins mCitrine and tagBFP on Cytex™ Aurora Flow Cytometry System (Cytex® Biosciences).

The expression of CAR constructs on the plasma membrane was analyzed with flow cytometry and Western blot by detection of genetically fused N-terminal Myc-tag. For flow cytometry experiments 5×10⁵ cells were collected 48h post electroporation and centrifuged for 5 min at 250 g. Pellet was washed with 1 mL PBS supplemented with 10% FBS and centrifuged for 5 min at 250 g. Pellet was resuspended in 80 uL FcR blocking reagent for 10 min at 4 °C. Antibodies against Myc-tag labeled with Alexa Fluor 647 were added in the ratio 1:100 for 30 min at 4 °C. After incubation cells were washed with addition of 1.8 mL PBS supplemented with 10% FBS and centrifuged for 5 min at 250 g. Pellet was resuspended in 300 uL PBS supplemented with 10% FBS. Fluorescence was detected with Cytex Aurora spectral flow cytometer. The rest of the cells were used for Western blot. Cells were collected and centrifuged for 5 min at 250 g. Pellet was washed with 1 mL PBS and centrifuged for 5 min at 250 g. Pellet was resuspended in 100 uL Ripa Lysis buffer supplemented with protease

inhibitor cocktail and incubated for 20 min at 4°C. After incubation cells were centrifuged for 15 min at 1700 g. Total protein concentration in supernatant was measured by Bicistronic acid method. Samples were complimented with denaturant SDS with reducing agent and incubated for 15 min at 95 °C. The total amount of proteins of 10 ug was loaded into Any kD™ Mini PROTEAN® TGX™ Precast Protein Gels with PageRuler™ Plus Prestained Protein Ladder (Thermo Scientific™) for control. SDS-PAGE was run under denaturizing conditions at 200 V for 35 min. Proteins were transferred to 0,45 um nitrocellulose membrane at 350 mA for 1.5 h. iBind (Thermo Scientific™) was used for antibody incubation. Myc-tagged CAR constructs were specifically detected with primary antibodies Rabbit anti-Myc-tag at 1:2000 and secondary antibodies Goat anti-rabbit-HRP at 1:3000 ratios. For loading control we detected Hsp70 protein using primary antibodies Mouse anti-Hsp70 at 1:1000 and secondary antibodies Goat anti-mouse-HRP at 1:3000 ratios. Detection of HRP was achieved by incubation of the membrane with SuperSignal™ West Femto Maximum Sensitivity Substrate (Thermo Scientific™), the loading control membrane part for 5 min, the CAR for 10min. The membrane was imaged with G:Box Chemi XT 4 Chemiluminescence and Fluorescence Imaging System (Syngene).”

Additional Supplementary Figure and its description:

Supplementary Fig. 10. Comparison of expression of CAR constructs. **a**, Schematic of 2nd generation antiCD19-CAR, antiCD20-CAR and dualCAR-T-4HB showing N-terminal Myc-tag. Jurkat cells were electroporated with 5 ug of construct(s) using Neon electroporation system. **b**, Flow cytometry analysis of CAR construct expression. 48h after electroporation 5×10^5 cells were collected and dyed with antiMyc-tag:AlexaFluor647 antibodies at 1:100 ratio. Presented is the comparison between (from left to right) empty pcDNA3 vector electroporated Jurkat cells (mock) 2nd generation

antiCD19-CAR, 2nd generation antiCD20-CAR and dualCAR-T-4HB. **b**, Western blot was performed from Jurkat cell, collected 48 h post electroporation. Cell lysates were prepared using Ripa buffer, supplemented with peptide inhibitors. Detection of Myc-tagged CAR constructs was performed using primary rabbit-antiMyc-tag antibodies at 1:2000 and secondary antibodies Goat anti-rabbit:HRP at 1:3000 ratio. Samples are as follows: (from left to right) empty pcDNA3 vector electroporated Jurkat cells (**1**), 2nd generation antiCD19-CAR (**2**), 2nd generation antiCD20-CAR (**3**) and dualCAR-T-4HB (**4**). Observed specific bands are accentuated with red arrows, at 55 kDa for 2nd generation CAR constructs, 46,8kDa for dualCAR-T-4HB constructs and 70kDa for loading control Hsp70.”

We renamed Supplemental figures following the new inserted supplemental figure accordingly and corrected the citation in the main text of the manuscript.

Unique capability: While the multimerization of 4HB is unique, there is no description of how their unique multimerization capability can lead to novel functions. Much of the applications shown in this work can be achieved with other more well-established dimerization domains, such as leucine zipper or PDZ domains. More descriptions that can delineate the novel function that can be created by using 4HB would be very helpful.

Response: The key advantage of 4HB modules is the ability to obtain several pairs from a single protein but above all the ability to combine more than 2 different partners, which is the limit for CC dimers and PDZ domains. Here we have demonstrated heterodimerization and heterotetramerization and regulation of the assembly of two partners by the presence of a regulatory peptide to regulate the assembly of BC +D (where each can be fused to a selected partner), by the addition of a peptide A, Fig. 6a).

Figure 4e time course is interesting, showing rapid activity. It would be interesting to know if the experiment would carry out longer, would the activity plateau.

Response: Longer duration of the experiment is shown in the Supplementary Fig 3d, which indeed shows a plateau after several hours.

Some of the RLU ranges from 0.5 (e.g., Figure 4g) to 150 (e.g., Figure 6a). Do they really have that much difference in RLU, or is it an artifact between different experiments? Since Luc is very sensitive, even a small level of expression can be detected. But it will not be very useful if the proteins are not expressed well. However, I do acknowledge that the dualCAR-T 4HB seems to work well.

Response: Large differences in RLU values mentioned by the reviewer are due to the use of different reporter systems in those experiments. In Fig. 6a we are monitoring the reconstitution of split Firefly luciferase directly fused to 4HB domains. Whereas in Fig. 4g we used a previously described protease reporter system, that includes a cyclic luciferase with a protease cleavage site. Where such reporters were used as indirect translation of 4HB reconstitution, a positive control (such as PPVp protease + cycLucPPVp in the same amount as the examined 4HB constructs) was used to determine the maximal values of the Firefly luciferase activity (see Supplementary Fig. 4b).

We hope that we appropriately responded to the issues raised by the reviewers and that you will find the revised manuscript appropriate for publication in *Nature Communications*.

With best regards,

Roman Jerala

Reviewers' Comments:

Reviewer #1:

Remarks to the Author:

Thank you for the work in manuscript. he manuscript has improved upon the revision process.

Reviewer #2:

Remarks to the Author:

Thank you for providing the response to my comments. I think the reply has addressed my comments adequately.